# Antibodies for Venezuelan Equine Encephalitis Virus Protect Embryoid Bodies from Chikungunya Virus

**DOI:** 10.3390/v12030262

**Published:** 2020-02-27

**Authors:** Emily M. Schultz, TyAnthony J. Jones, Kelli L. Barr

**Affiliations:** Department of Biology, Baylor University, Waco, TX 76798, USA; emily_schultz1@baylor.edu (E.M.S.); ty_jones1@baylor.edu (T.J.J.)

**Keywords:** chikungunya, congenital infection, antibody cross-reactivity, alphavirus

## Abstract

Chikungunya virus (CHIKV) is an alphavirus that causes febrile illness punctuated by severe polyarthralgia. After the emergence of CHIKV in the Western Hemisphere, multiple reports of congenital infections were published that documented neurological complications, cardiac defects, respiratory distress, and miscarriage. The Western Hemisphere is endemic to several alphaviruses, and whether antigenic cross-reactivity can impact the course of infection has not been explored. Recent advances in biomedical engineering have produced cell co-culture models that replicate the cellular interface at the maternal fetal axis. We employed a trans-well assay to determine if cross-reactive antibodies affected the movement and replication of CHIKV across placental cells and into an embryoid body. The data showed that antibodies to Venezuelan equine encephalitis virus significantly reduced CHIKV viral load in embryoid bodies. The data highlighted the fact that viral pathogenesis can be cell-specific and that exploiting antigenic cross-reactivity could be an avenue for reducing the impact of congenital CHIKV infections.

## 1. Introduction

Chikungunya virus (CHIKV) is an alphavirus vectored by *Aedes* mosquitos and is an enveloped + positive sense single stranded RNA ssRNA virus with a 12 kb genome. In humans, CHIKV can cause a febrile illness punctuated by severe polyarthralgia. Historically, CHIKV infection was thought to be self-limited, but increasing reports are showing that rheumatic and neurological sequelae can linger for years following infection [1,2,3]. When CHIKV emerged in the Western Hemisphere, neuroinvasive disease and congenital infections were reported at rates much higher than the Eastern Hemisphere [4]. CHIKV congenital infections were first reported during the 2005 outbreak on Reunion Island [5,6,7]. After the emergence of CHIKV in the Western Hemisphere, multiple reports of congenital infections were published that documented neurological complications, cardiac defects, respiratory distress, and miscarriage [5,8,9]. Although reports of congenital infection were published in many CHIKV-endemic locations, the vast majority originated from South and Central America [4].

South and Central America, along with the Caribbean, are endemic to several New World alphaviruses including Madariaga virus (MADV), Mayaro virus (MAYV), Eastern equine encephalitis (EEEV), Venezuelan equine encephalitis (VEEV), and Western equine encephalitis (WEEV). Human exposure prevalence for these viruses can be as high as 80% in some regions [10,11]. VEEV and CHIKV have been found to circulate in the same Central and South American regions [12,13,14,15]. For example, VEEV has been thought to be endemic in southern Mexico for decades, with seroprevalence rates between 18–75%, whereas the seroprevalence of CHIKV in southern Mexico has been determined by some studies to be as high as 85% [13,14]. Additionally, recent studies have indicated that antigenic cross-reactivity, antibody-mediated enhancement, and antibody cross-neutralization of alphaviruses can have a significant impact on the course of infection [16,17,18,19].

The study of congenital infections with these viruses is problematic, as most animal models do not reflect human disease. Although research has shown that non-human primates and sheep can serve as models, these systems can be expensive, labor intensive, and contain small sample sizes [20,21]. As a result, aside from *Zika virus*, there is limited research on other congenital arboviral infections outside of case reports and epidemiological studies. Recent advances in biomedical engineering have produced co-culture models using human stem cells or primary cells that replicate the cellular interface at the blood–brain barrier and the maternal fetal axis in order to evaluate the movement and effects of exogenous substances [22,23]. In order to determine if alphaviral antibody cross-reactivity could impact CHIKV congenital infections, we employed a trans-well assay to determine if cross-reactive antibodies impacted the pathogenesis or replication of CHIKV across placental cells and into an embryoid body.

Several in vitro models are used to measure translocation of products across the placenta, such as microfluidics, spheroids, organoids, and trans-well models [24,25,26]. Although scientists aim for model complexity in order to better replicate the placenta, the trans-well assay excels because it lends well to standardization and is easy to manipulate and reproduce [23,26]. For this study, we employed the trans-well approach due to its success in measuring the movement of nanoparticles across the placenta [27,28,29,30]. We followed Aengenheister et al. (2018) [27] by including an embryoid body (EB) in our co-culture model. EBs consist of self-organized stem cells that mimic the three dimensional structure of early peri-implantation development and possess the differentiation potential of early embryonic development [31]. We reasoned that this model would allow us to determine if virus and/or virus/antibody complexes could cross the placenta barrier and if they could impact an embryo.

We followed other research and used maternal syncytiotrophoblast and fetal umbilical vein cells (BeWo and HUVEC) cells for our placenta model [25,27,28] (Figure 1). BeWo cells are derived from human placenta and were used to represent syncytiotrophoblasts that form the placenta [32]. These cells form a continuous layer around the placenta, are in direct contact with the maternal blood supply, and function in nutrient exchange with the fetus [33]. These cells also form a barrier through which CHIKV must cross in order to invade the fetal environment. HUVEC cells are derived from the fetal umbilical vein and were cultured with a variety of factors to promote microvascular phenotypes (ATCC). Microvascular placental endothelial cells are involved with placental expansion during the first trimester and vascularization throughout pregnancy [33]. These cells are located adjacent to maternal syncytiotrophoblasts and maternal blood in the intervillous space in the placenta [33,34]. BeWo and HUVEC cells were plated on a permeable membrane that was coated with collagen to provide a basement membrane for cell attachment and for the development of microvillous structures as well as desmosome and tight junctions (Figure 1). Here, we describe the impact of VEEV antibodies on CHIKV infection using this in vitro congenital model.

## 2. Materials and Methods

### 2.1. Cell Culture and Virus Propagation

Primary human umbilical vein endothelial cells; normal, human, pooled (HUVEC) (ATCC PCS-100-013) were cultured in EndoGRO-MV-VEGF media (MilliporeSigma, Burlington, MA, USA) containing 5% fetal bovine serum (FBS). Additionally, human placental cells BeWo (ATCC CCL-98) were cultured in Ham’s F-12K (Kaighn’s) medium containing 10% FBS, penicillin/streptomycin, 1X non-essential amino acids, 1X Glutamax, and 1 mM HEPES. Lastly, *Cercopithecus aethiops* kidney cell line Vero E6 (ATCC CRL-1586) were grown in Dulbecco’s modified Eagle’s medium (DMEM) with 10% FBS, supplemented with penicillin/streptomycin, 1X non-essential amino acids, 1X Glutamax, and 1 mM HEPES. All cell lines were incubated at 37 °C/5% CO_2_. CHIKV (181/25) was obtained from BEI Resources (NR-50345) and expanded once in Vero cells. Polyclonal anti-Venezuelan equine encephalitis virus, TC-83 (subtype IA/B) glycoprotein (antiserum, goat), NR-9404, was obtained through BEI Resources (BEIresources.org) NIAID, NIH.

### 2.2. Embryoid Body Formation

Human-induced pluripotent stem cells (ATCC ACS-1019) were cultured in mTeSR1 media (StemCell Technologies, Vancouver, Canada) on plates coated with vitronectin XF (Stemcell Technologies). ACS-1019 cells were seeded in an AggreWell 400 24-well plate at a density of 2.4 × 10^5^ cells per well, following the manufacturer’s directions, in order to initiate EB formation (StemCell Technologies). ACS-1019 were cultured in the AggreWell microwells with AggreWell EB formation media for 72 h at 37 °C/5% CO_2_. After this, the resulting EBs were harvested and divided equally between replicates of each treatment.

### 2.3. Monolayer Infection and Imaging

Monolayers of BeWo and HUVEC cells were infected with 100 infectious units per well. After 48 h, samples were fixed with 4% paraformaldehyde and blocked with 5% lamb serum. Cells were stained with anti-CHIKV monoclonal antibody 3E7b and anti-MAP2 antibody (Novus Biologicals, Littleton, CO, USA). Slides were mounted with ProLong Gold Antifade Reagent with DAPI (Cell Signaling Technology, Danvers, MA, USA catalog #8961S) and images were obtained using an Olympus Fluoview 3000 confocal microscope. Images were processed using the Olympus Fluoview FV10-ASW 4.1 software package.

### 2.4. Trans-Well Co-Culture

Corning 12 mm Trans-well-COL collagen-coated 3.0 µm pore PTFE membrane insert (Corning, NY, USA catalog #3494) were seeded with HUVEC cells on the basolateral side of the insert at a concentration of 1.0 × 10^5^ cells per 200 µL, and BeWo cells were seeded on the apical side of the insert at a density of 1.5 × 10^5^ cells in 500 µL. The HUVEC monolayer on the basolateral side was achieved using methods described by Aengenheister et al. (2018) [27]. Briefly, inserts were inverted into 6-well plates, with 1 mL of phosphate buffered saline (PBS) in one well to ensure sufficient humidity. Rubber spacers (approximately 1.5 mm thick) were placed on the corner of the 6-well plate to lift up the lid slightly and prevent direct contact of the lid with the inverted insert. After the basolateral side was seeded with HUVECs and the lid was replaced, there was slight adhesion between the lid and the media. HUVEC-seeded inserts were then incubated at 37 °C/5% CO2 for 2 h, and afterwards the inserts were placed back into the 12 well plate containing fresh HUVEC media. Co-cultures were incubated for 72 h with the media being changed every 48 h until a 100% confluent layer was observed.

### 2.5. Trans-Well Neutralization Assay

Prior to infection, the media in each basolateral well was replaced with 1/2 HUVEC media 1/2 EB formation media. EBs were added to the bottom of well (Figure 1). Neutralization assays using VEEV serum were performed using a 1:200 dilution of serum in PBS. A total of 10,000 infectious units of virus in PBS were incubated with serum for 1 h at 37 °C, after which BeWo cells, apical side of the trans-well inserts, were inoculated with the mixture. Assay controls included treatments of mock infection and virus only. Culture supernatant (BeWo and HUVEC) and EB samples were taken at 24, 48, and 72 h. EBs were separated from culture supernatant by centrifugation at 400× *g* for 4 min. The supernatant was aspirated. EBs were rinsed in 1 mL PBS and centrifuged again at 400× *g* for 4 min. The supernatant was removed, and the EBs were resuspended in PBS and homogenized by vigorous trituration. Results are expressed as an average between two independent trials with three replicates for each treatment. Pairwise comparisons were performed between relative treatments using Student’s *t*-test with a Tukey post-hoc test.

### 2.6. Viral Quantification

Plaque assays were performed using culture supernatant from the HUVEC and BeWo monolayers, as well as using the pooled supernatant samples from each treatment at each time point taken during the course of the trans-well experiment following a method described previously [35]. EBs were separated as described above. Briefly, serial dilutions of culture supernatant or EBs in PBS were inoculated onto confluent Vero E6 cells and covered with 0.25% methylcellulose overlay. After 3 days, the overlay was removed, and cells were stained with Coomassie blue. Viral RNA was extracted using a kit in accordance to the manufacturer’s instructions (Zymo quick viral RNA kit). Quantitative real-time PCR, performed on all supernatant samples taken from all treatments at each time point using Verso One-Step RT-qPCR Kit, SYBR Green, ROX (Thermo Fisher), and primers designed by Patel et al. (2019), which are specific for the CHIKV E1 gene [36]. A non-template control was used to normalize the RT-qPCR results. Pairwise comparisons between treatments were performed using raw Ct values with Student’s *t*-test with a Tukey post-hoc test. Relative fold change was calculated via the ∆∆Ct method using the non-infected reference cell line as a baseline.

## 3. Results

### 3.1. Chikungunya Infected Maternal and Fetal Placental Cells

Both BeWo and HUVEC cell lines were infected with CHIKV and visualized using anti-CHIKV monoclonal antibody 3E7b (EMD Millipore). The images show that both BeWo and HUVEC cells were permissive to CHIKV infection with significantly more fluorescence detected in CHIKV-infected cells than in their non-infected controls (Figure 2). BeWo monolayers exhibited no noticeable cytopathic effects (CPE) At 72 h post infection (p.i.), HUVEC cells showed increased CPE consisting of cell rounding and sloughing, whereas BeWo cells showed no CPE.

### 3.2. Trans-Well Neutralization Assay

CHIKV was detected via viral plaque assay and RT-PCR 24 h p.i. and continuing through 72 h p.i. in BeWo and HUVEC cells. Pairwise comparisons of timepoints of CHIKV-infected BeWo cells indicated a significant rise in viral titers at 72 h compared with 24 h p.i. (*p* = 0.0387). The presence of VEEV antibodies resulted in a significant reduction in viral plaques at 72 h post-inoculation in BeWo cells when compared to CHIKV-infected BeWo cells without VEEV antibodies (*p* = 0.0162) (Figure 3A). When evaluated by RT-PCR, significant increases in viral genome copy were observed over all time points for BeWo cells for both CHIKV only (*p* = 0.021–0.037) and CHIKV + VEEV antibody (*p* = 0.009–0.021) treatments (Figure 3B). Pairwise comparisons between CHIKV only and CHIKV + VEEV antibodies at each time point also indicted no significant changes in Ct value when VEEV antibodies were present in BeWo cells (Figure 3B).

HIKV was detected in HUVEC cells with significant increase in viral titer at 48 h compared to 24 h (*p* = 0.0372), followed by a significant reduction in viral titer at 72 h compared to 48 h (*p* = 0.0219) (Figure 3C). The presence of VEEV antibodies did not result in a significant reduction in viral plaques at any time point in HUVEC cells when compared to CHIKV-infected HUVEC cells without VEEV antibodies. When evaluated by RT-PCR, significant increases in viral genome copy were observed at all time points for HUVEC cells for both CHIKV only (*p* = 0.021–0.037) and CHIKV + VEEV antibody (*p* = 0.009–0.021) treatments (Figure 3D). Pairwise comparisons between CHIKV only and CHIKV + VEEV antibodies at each time point indicted that no significant changes in Ct value were found when VEEV antibodies were present in HUVEC cells (Figure 3D). Because cell culture supernatant was used, the discrepancies between the quantities of infectious virus and nucleic acid were likely a reflection on the proportion of defective virions that exited from the host cell.

For EBs, CHIKV was detected by viral plaque assay at 24, 48, and 72 h p.i. (Figure 4). CHIKV was also detected in EBs when VEEV antibody was present at 24, 48, and 72 h p.i. (Figure 4). When VEEV antibodies were present, significantly fewer viral plaques were measured at 72 h p.i. when compared to the virus-only treatment at the same time point (*p* = 0.027) (Figure 4A). Conversely, when evaluated by RT-PCR at 72 h p.i., viral genome copies were detected at a 5-fold greater quantity for CHIKV alone and a 29.95-fold greater quantity for CHIKV + VEEV antibodies. These were significantly more than at the previous time points (*p* = 0.002). Additionally, at 48 h p.i., the presence of VEEV antibodies resulted in a significant decrease in CHIKV genomic material as detected by RT-RCR (*p* = 0.0103). Furthermore, at 72 h p.i., CHIKV was detected at significantly higher levels by RT-PCR when VEEV antibodies were present as opposed to CHIKV alone at 72 h (*p* = 0.0267) (Figure 4B). The discrepancies between the quantities of infectious virus and genome copy we observed could have been a reflection of viral assembly on the proportion of infectious virions that exited from the host cell.

## 4. Discussion

The data showed that CHIKV can replicate in maternal and fetal placental cells as well as in EBs. Previous work has shown that the ability for CHIKV to infect and replicate in specific cells types is host-specific as has been shown for multiple flaviviruses [37,38,39]. Thus, it is necessary to identify which cells participate in viral pathogenesis. Reports on CHIKV replication efficiency in mammals and cell lines have shown a range of detection depending on the host [40,41]. The data here show that in BeWo and HUVEC cells, CHIKV replication peaked at 48 h p.i., whereas in EBs, CHIKV replication was greatest at 72 h p.i. Whether this 72 h peak reflected an infection delay caused by the movement of CHIKV through two cell monolayers and a basement membrane requires further investigation.

CHIKV has been detected in the placenta and amniotic fluid of infected mothers, which supports this platform as an in vitro mechanism for studying the kinetics of congenital CHIKV infections [5,6,42,43]. The virus was quantified via titration in Vero cells and by RT-PCR. Titration quantifies infectious units whereas RT-PCR measures genome copies. In this study, there was little correlation between infectious units and genome copies. Further, it was found that significantly more genome copies were produced in EBs when VEEV antibodies were present. This might suggest potential antibody-mediated enhancement. Although this is commonly associated with flaviviruses, antibody-mediated enhancement has also been documented for two alphaviruses, Ross River virus (RRV), and recently, CHIKV (by RT-PCR) [16,44]. These studies measured the enhancement of sub-neutralizing RRV or CHIKV antibodies against subsequent infections with the same virus. However, outside these two studies, antibody-mediated cross enhancement has not been explored in depth for other alphaviruses. Regardless, viral plaque assays in this study showed that infection of EBs was significantly reduced when VEEV antibodies were present, suggesting that these antibodies could be interfering with viral assembly, maturation, or exit. Recent work has shown that neutralizing antibodies interact with viral glycoproteins present on the cell surface, which inhibits the budding of CHIKV at the plasma membrane [45,46,47]. Furthermore, CHIKV monoclonal antibodies have been shown to interact with viral envelope proteins of other alphaviruses to inhibit both at viral fusion and exit [47].

The data show that CHIKV actively replicates in both cell lines by 48 h p.i. Whether infection of the basolateral side of the membrane was due to cell–cell contact or virus escape into the basolateral media is not known. Although the detection of CHIKV in cells located on either side of the maternal fetal axis in this study supports case reports of CHIKV congenital infections and isolation of CHIKV from placentas, the susceptibility observed in this study may not reflect the cellular tropism of CHIKV in actual placentas [5,6,42,48].

A major limitation of this study was the omission of Hofbauer cells from the model. These cells function as the antigen presenting macrophages in the placental villous stroma and function in host defense [33]. Although the role of Hofbauer cells has not been described for CHIKV, several reports document their role for other congenital viral infections [49,50,51,52,53,54]. Because Hofbauer cells are antigen-presenting, they could play a role in viral enhancement, as has been reported for *Zika virus* [50].

Cross-protective antibodies are commonly targeted for their use in vaccines and therapeutics, and antigenic cross-reactivity has been described for CHIKV [47]. This study found that VEEV antibody-mediated neutralization of CHIKV occurred in BeWo cells and EBs. The polyclonal serum used in this study was pooled from several goats obtained beginning 2 weeks post final inoculation and continuing for at least 4 weeks. The development of neutralizing antibodies begins about 2 weeks p.i. and continues to rise over the next 3–6 months. Thus, the serum used here may not have represented the full repertoire of VEEV neutralizing or enhancing antibodies. Furthermore, the serum used in this study was from goats vaccinated with TC-83 VEEV vaccine due to its availability from BEI Resources and its defined nature. However, the response of CHIKV to VEEV antibodies may not represent the full potential of CHIKV cross-reactivity in the New World.

Although alphaviruses possess a high degree of genetic diversity, phylogenetic studies have shown distinct groupings of Old and New World alphaviruses [55]. The VEEV complex itself contains a high degree of genetic and antigenic variation, and human seroprevalence rates have been reported to range from 14% to 33% [55,56,57,58]. Other New World alphaviruses such as Western Equine Encephalitis and Eastern Equine Encephalitis groups have less genetic diversity than VEEV and are distributed over larger geographic areas than viruses in the VEEV complex [55]. Further, MAYV and MADV are emerging as new threats to human health. MAYV belongs to the Semliki Forest Virus complex, which also contains CHIKV and could possess more antigenic cross-reactivity than VEEV [55]. Clearly this area warrants further investigation.

## 5. Conclusions

Early immune VEEV antibodies significantly reduce CHIKV in BeWo and EBs 72 h p.i. There may be other cross-neutralizing antibodies from other alphaviruses that also impact congenital CHIKV infections. More information on CHIKV congenital infections is needed, as is work that evaluates other alphaviruses and Hofbauer cells.

## Figures and Tables

**Figure 1 viruses-12-00262-f001:**
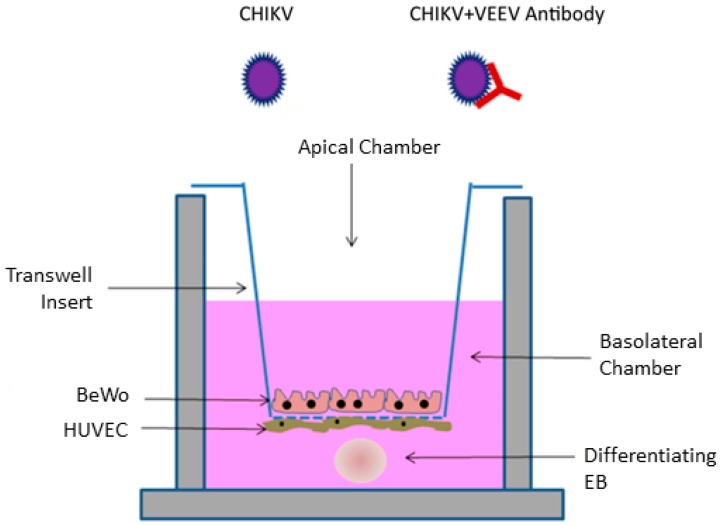
Experimental model of the trans-well co-culture assay modified from Campagnolo et al. (2018) [23]. Co-cultures of BeWo, HUVEC, and embryoid body (EB) were apically infected with either Chikungunya virus (CHIKV) or CHIKV + VEEV (Venezuelan equine encephalitis) antibody.

**Figure 2 viruses-12-00262-f002:**
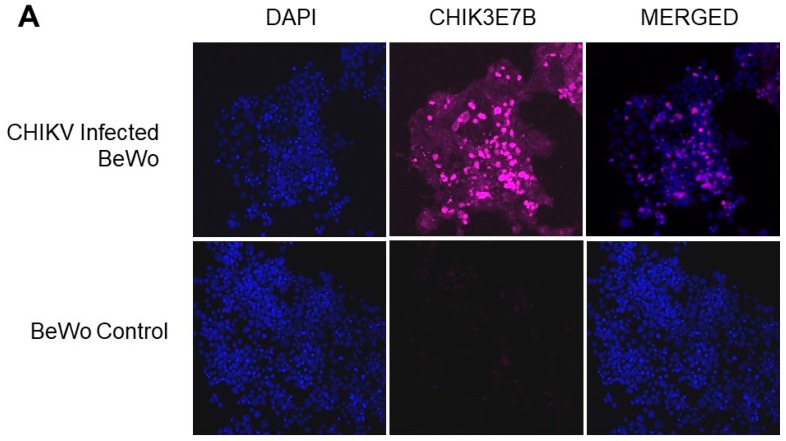
CHIKV infection in BeWo and HUVEC cells. (**A**) BeWo cells were infected with CHIKV and stained at 48 h p.i. (**B**) HUVEC cells were infected with CHIKV and stained at 72 h p.i. (blue = DAPI, pink = CHIK3E7b).

**Figure 3 viruses-12-00262-f003:**
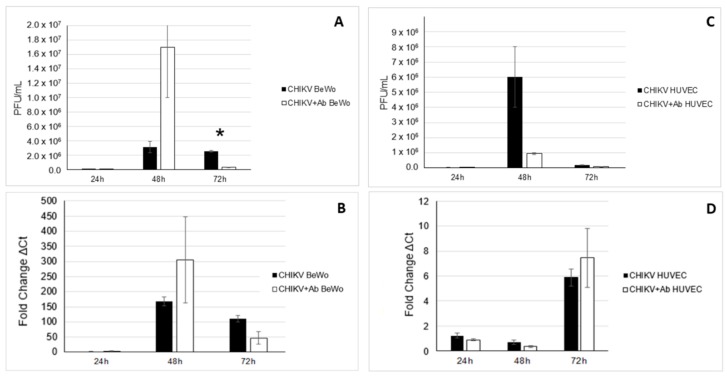
Viral titration as well as RT-PCR of BeWo and HUVEC cell lines. Viral titrations via plaque assay were performed for BeWo and HUVEC cells infected with CHIKV and CHIKV + VEEV antibodies. (**A**) Viral titrations showed that significantly less CHIKV was detected in BeWo cells at 72 h when VEEV antibodies were present (*p* = 0.0162, *n* = 6). (**B**) Significant increases in viral genome copy were observed at all time points for BeWo cells for both CHIKV only (*p* = 0.021–0.037) and CHIKV + VEEV antibody (*p* = 0.009–0.021) treatments. However, there was no significant difference in Ct value at each time point for pairwise comparison for CHIKV only compared to CHIKV + VEEV antibodies. (**C**) Viral titrations displayed that there was no significant difference in CHIKV at any time point, with or without the presence of VEEV antibodies in HUVEC cells. There was significant increase in viral titers of CHIKV between 24 and 48 h (*p* = 0.0372, *n* = 6), but a significant decrease in viral titers between 48 and 72 h (*p* = 0.029, *n* = 6). (**D**) HUVEC cells exhibited significant increases in viral genome copies over all time points for CHIKV only (*p* = 0.021–0.037, *n* = 6) and CHIKV + VEEV antibodies (*p* = 0.009–0.021, *n* = 6). * denotes statistical significance between treatments.

**Figure 4 viruses-12-00262-f004:**
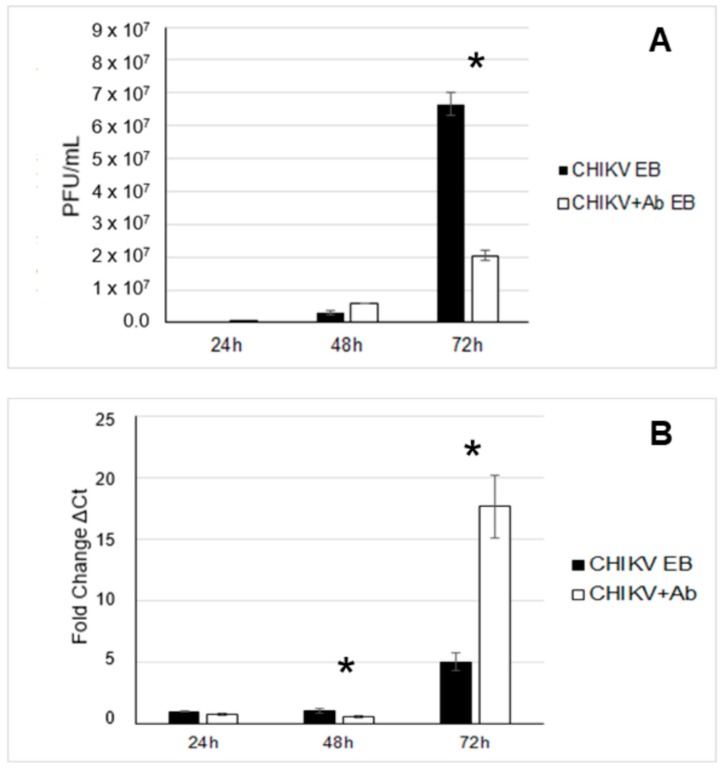
Detection of CHIKV in EBs. (**A**) Viral titration of EBs was performed via plaque assay. Significantly less CHIKV was detected at 72 h p.i. when VEEV antibodies were present (*p* = 0.027, *n* = 6). (**B**) RT-PCR detected significantly more genome copies at 72 h p.i. when VEEV antibodies were present (*p* = 0.0267, *n* = 6), and significantly less at 48 h when VEEV antibodies were present (*p* = 0.0103, *n* = 6). * denotes statistical significance between treatments.

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
