# Peer review of "Antibodies for Venezuelan Equine Encephalitis Virus Protect Embryoid Bodies from Chikungunya Virus"

_viruses, 2020, doi:10.3390/v12030262_

Round 1

Reviewer 1 Report

The manuscript and description of methods have been improved. Fig 2B is still not convincing. Better pictures should be shown. Why we do not see the DAPI and MAP staining?

Author Response

Fig 2B is still not convincing. Better pictures should be shown.

We have generated high resolution figures.

Why we do not see the DAPI and MAP staining?

We removed the MAP2 staining. We think you refer to the CHIKV infected huvecs with no DAPI. DAPI staining is present in the CHIKV-infected cells. There are fewer cells due to viral CPE to stain which is why the DAPI looks so reduced. We have generated higher resolution figures.

Reviewer 2 Report

In the manuscript of “Antibodies for Venezuelan Equine Encephalitis Virus Protect Embryoid Bodies from Chikungunya Virus”, the authors employed a trans-well assay to determine if cross-reactive antibodies affected the movement and replication of CHIKV across placental cells and into an embryoid body. The authors tried to address an interesting question and the results from the study should provide valuable information about the effect of cross-reactive anti-alphavirus antibodies, especially multiple different alphaviruses are circulating in some areas. However the study is weakened by the lack of details of methods, missing experimental controls, and unclear presentation of results.

Major comments:

  1. How did the authors measure the virus production by different cells in the tans-well neutralization assay? If the authors collected the supernatant and did plaque assay, how did they distinguish virus produced from HUVEC and EB? In the trans-well setting, HUVEC and EB were cultured in the same media.

  1. How did the authors quantify the virus by qRT-PCR? If the viral RNA was extracted from the culture supernatant, then the discrepancies between the infectious units and vRNA copies shouldn’t relate to viral assembly and exit as the authors claimed in Page 6 line 200-202.

  1. Control experiments demonstrating the anti-VEEV function of VEEV antibodies used in the study are missing. Control antibodies are missing for the trans-well neutralization assay.

  1. The authors need to address why anti-VEEV antibodies slightly reduced infectious virus (<3-fold) but increased vRNA copies from EB (Figure 4). Why log scale was used for figure 3A and 3C, but linear scale was used for figure 4A.

Minor comments:

  1. The authors often claimed “more” or “increase” without indicating what they were comparing. For example, page 5 line 167, “The images show that … with significantly more fluorescence detected in CHIKV-infected cells.” Page 6, line 185, When evaluated by RT-PCR, significant increases in viral genome copy were observed over all time points for BeWo cells for both CHIKV only and CHIKV+VEEV antibody”, et al.

  1. Figure 2, it seems that BeWo cells are more susceptible to CHIKV than HUVEC cells as the CHIKV infected BeWo cells expressed higher level of viral antigens than CHIKV infected HUVEC cells. Due to the dramatic difference of viral antigen expression levels between two cell lines, the authors shouldn’t claim more paranuclear staining in BeWo than in HUVEC as in page 5 line 171.

Author Response

How did the authors measure the virus production by different cells in the tans-well neutralization assay? If the authors collected the supernatant and did plaque assay, how did they distinguish virus produced from HUVEC and EB? In the trans-well setting, HUVEC and EB were cultured in the same media.

    1. Section 2.5 of the manuscript describes how we were able to distinguish EB viral RNA from HUVEC viral RNA. In short, supernatant was removed from the EBs via washing with PBS and centrifugation.      

How did the authors quantify the virus by qRT-PCR? If the viral RNA was extracted from the culture supernatant, then the discrepancies between the infectious units and vRNA copies shouldn’t relate to viral assembly and exit as the authors claimed in Page 6 line 200-202.

    1. Quantification of viral RNA via RT-PCR is described in section 2.6. While infectious virus was not quantified per se.       We measured fold change using the ∆∆Ct method using the non-infected reference cell line as a baseline. A standard curve quantification strategy would not have been appropriate due to the variable permissiveness between cell types. Viral RNA was detected via supernatant from BeWo and Huvec cells and directly from Ebs.

    1. We agree with your statement and have changed the text to reflect infectious verses non-infectious particles.       The text has been edited and now reads: “Since cell culture supernatant was used, the discrepancies between the quantities of infectious virus and nucleic acid are likely a reflection on the proportion of defective virions that exited from the host cell.”

Control experiments demonstrating the anti-VEEV function of VEEV antibodies used in the study are missing. Control antibodies are missing for the trans-well neutralization assay.For all experiments a mock infection (media only) was used as a negative control. The positive control included the CHIKV-only infections. These are the standard controls used in neutralization/enhancement assays when the activity of polyclonal serum and monoclonal antibodies are being evaluated (Pal et al. 2013 PMID 23637602 and Quiroz et al 2019 PMID 31697791 and Chua et al 2016 PMID: 27571254)  

  1. Unfortunately, polyclonal CHIKV-neutralizing serum is not publicly available and CHIKV monoclonal antibodies that are available for purchase have no documented activity against the strain of CHIKV used in this study. Further, the only publicly available VEEV polyclonal serum was used as the experimental variable in this study and thus, could not be used as a control.

The authors need to address why anti-VEEV antibodies slightly reduced infectious virus (<3-fold) but increased vRNA copies from EB (Figure 4). Why log scale was used for figure 3A and 3C, but linear scale was used for figure 4A.

This is explained at lines 230-232: The discrepancies between the quantities of infectious virus and genome copy we observed could be a reflection viral assembly on the proportion of infectious virions that exited from the host cell.Minor comments:

  1. The authors often claimed “more” or “increase” without indicating what they were comparing. For example, page 5 line 167, “The images show that … with significantly more fluorescence detected in CHIKV-infected cells.” Page 6, line 185, When evaluated by RT-PCR, significant increases in viral genome copy were observed over all time points for BeWo cells for both CHIKV only and CHIKV+VEEV antibody”, et al.

We have made edits throughout the text clarifying comparisons per your suggestion.

Line 167

Line 180

  1. Figure 2, it seems that BeWo cells are more susceptible to CHIKV than HUVEC cells as the CHIKV infected BeWo cells expressed higher level of viral antigens than CHIKV infected HUVEC cells. Due to the dramatic difference of viral antigen expression levels between two cell lines, the authors shouldn’t claim more paranuclear staining in BeWo than in HUVEC as in page 5 line 171.
    1. We deleted this claim per your suggestion.

Round 2

Reviewer 2 Report

Line 268-270, the authors suggested "Regardless, viral plaque assays in this study showed that infection of EBs was significantly reduced when VEEV antibodies were present, suggesting that these antibodies could be interfering with viral assembly, maturation, or exit." Literature on antibody mediated inhibition of CHIKV budding should be included. (Cell Host Microbe. 2018 Sep 12;24(3):417-428; Cell Rep. 2015 Dec 22;13(11):2553-2564; Cell. 2015 Nov 19;163(5):1095-1107)

Otherwise, the authors addressed all the comments.

Author Response

Thank you for those references! 

We have modified the manuscript per your request and have added the following lines 265-269 using the suggested citations.

"Recent work has shown that neutralizing antibodies interact with viral glycoproteins present on the cell surface which inhibits the budding of CHIKV at the plasma membrane (45-47). Furthermore, CHIKV monoclonal antibodies have been shown to interact with viral envelope proteins of other alphaviruses to inhibit both at viral fusion and exit (47)."

This manuscript is a resubmission of an earlier submission. The following is a list of the peer review reports and author responses from that submission.

Round 1

Reviewer 1 Report

In this scientific report, the authors investigate an interesting aspect of viral pathogenesis that is becoming important. Indeed alternative route of transmission of arboviruses, such as vertical transmission, and the cross reactivity with antibodies of co-circulating viruses may affect viruses spreading and pathogenesis.

The model the authors chose to reproduce maternal-fetal axis it is intriguing but may have some limitations that should be addressed in the discussion section. The authors point at the lack of hofbauer cells in their system but also the tropism of the virus for BeWo and HUVEC cells may not reflect the actual tropism of CHIKV for placental cells, as it was demonstrated for ZIKV. Moreover, the authors describe the BeWo as cells of maternal origin, which is not the case (syncytiotrophoblast are of fetal origin and male cells…).

It could also be fine to discuss the choice of this particular system which resemble a placenta at term since cytotrophoblast are lacking. The authors could include in the introduction references on vertical transmission of CHIKV suggesting a mechanisms and deriving from in vitro, animal model, and epidemiological studies.

See: Couderc T., et al. A mouse model for Chikungunya: Young age and inefficient type-I interferon signaling are risk factors for severe disease. PLoS Pathog. 2008; Gérardin P., et al. Multidisciplinary Prospective Study of Mother-to-Child Chikungunya Virus Infections on the Island of La Réunion. PLoS Med. 2008; Touret Y.,et al. Early maternal-fetal transmission of the Chikungunya virus. Presse Med. 2006; Charlier C., et al. Arboviruses and pregnancy: Maternal, fetal, and neonatal effects. Lancet Child Adolesc. Heal. 2017; Matusali G., et al. Tropism of the chikungunya virus. Viruses 2019

In introduction section: line 44 correct the sentence and line 47 when speaking of new models,the authors should cite organoid systems.

In material section: When in the experimental protocol do the authors use Humani iPS? If they use it for embryoid bodies formation I suggest to move this line to the next material section. Line 67 please add information on how you obtain embryoid bodies and culture protocol. line 73-74: correct in: cells were stained with anti-CHIKV monoclonal antibody 3E7b and anti-MAP2 antibody.

Fig.1 correct the reference in bibliography section

As for presentation of results, the graphs could be improved, we do not really understand at which pairs the statistic bars refer to. Moreover, Fig 3 and 4 could be merged in one figure and presented differently:

4A should be 3B; 3B become 3C, 4B become 3D. Section 3.2 should be named neutralization assay or point at the findings. The authors should explain why they stain MAP2, and choose better picture for showing HUVEC infection by CHIKV, the IF picture they show it is not convincing (fig 2B). In addition,: line 126-128 it is not clear the choice to use a statistical test for fluorescence when comparing not infected to infected cells.

Section on EB bodies infection has no title. line 171-172 this is a confusing sentence, please rephrase.

Discussion section: line 196-198 not clear and not convincing, Moreover, this point is discussed later.

Importantly, the authors should better discuss the discordant findings by RT-PCR and plaque assay.

It would be interesting to read a sentence on the specific areas of co-circulation of VEEV and CHIKV.

Finally statistical tests and number of experiments (n=) should be described for each findings in figure legends.

Reviewer 2 Report

The authors aim to evaluate the ability of antibodies against VEEV to alter CHIKV infectivity.  The authors evaluated the impact of VEEV antibodies on CHIKV replication and viral RNA production in HUVECs, BeWo cells and embryoid bodies (EBs) in vitro. Monolayer cultures of HUVECs and BeWo cells showed no impact on CHIKV replication with the addition of VEEV antibodies.  However, transwell co-cultures of HUVECs, BeWo cells and EBs showed a reduction of CHIKV infectious titers, but an increase in CHIKV viral RNA in the presence of VEEV antibodies. 

Unfortunately, the results are poorly controlled and the experimental system was not defined/validated.  Moreover, the data are all descriptive with little to no mechanistic information regarding how the inhibition is occurring.

Major comments

Figure 2: The MAP2 and CHIKV staining for the HUVECs is not convincing. Given that the CHIKV is replicating quite well in HUVECs (according to Figure 3), the authors should provide better images.

More information about the EBs is needed. 

There are no experiments validating and characterizing the experimental system show in Figure 1. 

Negative and positive control antibodies are needed for all experiments.

The statistical significance differences on the graphs are not clear (which comparisons are being made?).

The authors need to explain how HUVECs only have 2-fold viral RNA at 48 hours (Figure 4B), but ~10^7 pfu/ml virus released (Figure 3B)?

The experimental set-up in Figure 5 needs more description.  Was the set-up in Figure 1 used for this experiment?  How can the authors determine that the infectious virus was from the EBs?  Was the viral RNA from the sups or from the cells? If the viral RNA was from the cells, then to further support the idea that CHIKV was neutralized by the VEEV Abs, the amount of viral RNA in the sups should also be determined.

The antibody used was serum from goats.  This limits the interpretation of the results. 

Minor comments

Line 44 needs to be rewritten to more fully explain the limitations of the larger animal models.

Figure 2: Define MAP2

Figure 3 and Figure 5A: please change the y-axis labels to pfu/mL and display the values using scientific notation.

Figures 4 and 5: Please alter the y-axis values to not show any decimal places.

Line 107-108: The authors indicated: “Percent neutralization was calculated using the virus only treatment as a baseline”. No graphs show percent neutralization.  Where are these data shown?

Line 196-197: What do the authors mean by “significantly more copies of the envelope gene were produced than infectious particles by the EBs”?  The envelope gene detected can be part of the genomic RNA, the negative strand RNA, or the subgenomic RNA. 

Line 199-200 indicates “BeWo cells are derived from human placenta and were used to represent syncytiotrophoblasts which form the placenta”. This statement needs to come earlier in the manuscript to explain the experimental system.  It would also be helpful if the cells used would be indicated in Figure 1.

Was a house keeping gene used for the RT-qPCR analysis?  If not, the authors need to indicate why one wasn’t used.